# Listening to Their Nights: Sleep Disruptions in Captive Housed Chimpanzees Affect Their Daytime Behavior

**DOI:** 10.3390/ani13040696

**Published:** 2023-02-16

**Authors:** Pablo R. Ayuso, Olga Feliu, David Riba, Dietmar Crailsheim

**Affiliations:** 1Research Department, Fundació Mona, 17457 Girona, Spain; 2Department of Clinical Psychology and Psychobiology, Faculty of Psychology, University of Barcelona, 08035 Barcelona, Spain; 3Facultat de Lletres, Universitat de Girona, 17003 Girona, Spain

**Keywords:** chimpanzee, nocturnal activity, sleep disruption, sound recording, vocalization, sanctuary, welfare, agonistic behavior, temperature, humidity

## Abstract

**Simple Summary:**

Ensuring proper sleep is essential for the health of all animals, including chimpanzees. Many studies have demonstrated that sleep plays a crucial role in physical and behavioral regulation, and sleep disruptions may cause negative consequences for both. This study explores how environmental conditions and nocturnal disturbances affect the sleep and daytime behavior of sanctuary-housed chimpanzees, respectively. Our results indicate that indoor temperature and humidity affect chimpanzee nocturnal activity. Furthermore, the frequency and intensity of nocturnal disturbances influence the chimpanzee’s behavior the following day, i.e., time spent in inactivity, abnormal, self-directed, and affiliative behaviors. These findings highlight the importance of controlling factors that affect the quality of chimpanzees’ sleep at night. The use of affordable audio recordings has the potential to provide valuable information regarding the chimpanzee’s nocturnal activity and wellbeing.

**Abstract:**

Chimpanzee’s nocturnal sleep is a dynamic and complex process, still not fully understood. As in humans, not getting enough quality sleep due to frequent or lasting disruptions may affect their physical and mental health, hence wellbeing, which may be reflected in their daytime behavior. This study aims to understand the impact of abiotic factors, such as temperature and humidity on the nocturnal activity as well as the impact of nocturnal awakening events on daytime behavior in sanctuary-housed chimpanzees. We monitored noisy nocturnal activity through audio recordings for one year, documenting the number and duration of sound events produced by chimpanzees to indicate sleep fragmentation and disruption intensity, respectively. Our results indicate that indoor temperature and humidity indeed influence the chimpanzee’s nocturnal activity. Furthermore, sleep fragmentation and intensity of nocturnal events significantly influenced the following day’s behavior. After nights marked by frequent and/or intense sleep disruptions, higher levels of inactivity, and abnormal and self-directed behaviors were observed, and chimpanzees spent more time on affiliative interactions and in social proximity. These findings highlight the importance of controlling factors influencing nocturnal sleep quality. Furthermore, we demonstrated that economic audio recordings used to monitor nocturnal activity, provide insights into the chimpanzee’s behavior and wellbeing.

## 1. Introduction

Ideal sleep patterns and the consequences of not achieving enough quality sleep have been extensively studied in humans. Sleep deprivation, insufficient sleep, and sleep fragmentation have been linked to poor health and have been demonstrated to lead to a lack of attention, reduced concentration, reduced motor functions, impairment of emotion regulation as well as the development of diseases, depression, and morbidity, hence negatively impacting an individual’s wellbeing [1,2,3,4]. Although far less studied, sleep is suggested to be an equally crucial component for the health and wellbeing of non-human primates and other animals as well [5,6,7,8,9,10,11,12]. Adult chimpanzees in captivity have been reported to sleep an average of 8.8–12 h per day, with 90-min sleep cycles and 2 h of REM sleep [13,14,15,16]. In order to acquire conditions favoring quality sleep, wild-ranging chimpanzees tend to build nests in trees [17]. Nest building in great apes is suggested to serve several purposes, such as to increase comfort considering their larger body size; as a thermoregulatory strategy in extreme environments; to reduce the risk of disease transmission and as a defense mechanism against predators, i.e., a mechanism allowing to lessen a certain amount of alertness [18]. Furthermore, several studies were able to discern social dynamics based on the proximity and distribution of nest locations within chimpanzee party’s sleeping sites [5,6,19,20]. The freedom to choose which community members are nesting in proximity and which group members to avoid might also allow chimpanzees to establish a social environment reducing intra-group tensions, i.e., diminishing the need to be overly vigilant towards group members.

As diurnal animals, chimpanzees are typically thought to be inactive at night, resting and sleeping for most of the time. However, both in the wild and in captivity, chimpanzees have been observed to exhibit a significant amount of nighttime activity [10,21]. Some nocturnal movements may simply be brief changes in posture, but other reasons for nocturnal restlessness may include alertness to predators or other environmental disturbances [6]. Chimpanzees’ sleep may also be interrupted by the activity of group members, which may involve changes in posture, defecations, or vocalizations such as pant hoots [22]. However, in the wild, social interactions between group members have been rarely observed at night [23], except for interactions between mothers and their offspring and occasional sightings of behaviors related to mating [6]. Apart from disruptions caused by conspecifics, other factors such as environmental and physiological factors may cause chimpanzees to wake up during the night. This includes sleep interruptions produced by the activity of nocturnal arboreal animals [21]. Chimpanzees have also been observed to travel and forage [10] or raid crops at night [9]. This activity is more likely occurring in seasons marked by higher daytime heat stress, in hot and dry environments [10,23,24]; when moonlight provides increased visibility [10], and has been argued to be a mechanism to avoid human presence [9].

In captivity, both the physical environment and factors potentially impacting sleep patterns differ greatly when compared to the wild. Chimpanzees in captivity are typically restrained to indoor facilities at night. Such facilities tend to be much smaller in comparison to outdoor enclosures used during the day. Furthermore, chimpanzees in captivity often lack control and choice regarding with whom to share night quarters. Additionally, organizations may be taking care of more than one group of chimpanzees, often housed within the same building or close by. Hence, chimpanzees from different groups might not be able to see each other but are likely to hear each other’s vocalizations and any noisy behaviors at night. Thus, individuals that are restless at night might not only disrupt the sleep of chimpanzees residing within the same quarters but potentially also affect chimpanzees in neighboring night quarters and/or chimpanzees from other groups. Baker and Aureli [25] as well as Videan et al. [26] found vocalizations and noisy displays produced by neighboring chimpanzees’ groups in captivity to produce a behavioral contagion. Specifically, they found chimpanzees to exhibit higher rates of vocalization, display, and agonistic behaviors when levels of neighbor vocalization were labeled as high. Furthermore, many care institution report issues with nocturnal aggressions and wounding incidence in captive-housed chimpanzees [20]. Typically, these are argued to be at least partially produced by the limited size of indoor enclosures, not ideal group compositions, and/or the chimpanzee’s disability to avoid specific group members. The potential occurrence and likelihood of agonistic nocturnal interactions are expected to produce heightened alertness toward group members residing within the same night quarters. Although sleep in wild and captive chimpanzees tends to be frequently interrupted, it is likely that an increase in the number and intensity of these nocturnal disturbances may produce adverse effects on their physical and mental health, as well as their daytime behavior, as has been extensively studied in humans [7,11,27] and other non-human species [28,29,30]. Studies conducted on humans and laboratory animals demonstrated acute sleep deprivation to negatively affect mood and emotional wellbeing, as well as cognitive performance and motor functions. Chronic sleep restriction can lead to significant impairment of emotional and behavioral wellbeing [31] and may have adverse effects on memory consolidation and learning capacity [7], with insomnia being an important predictor of depression [32,33]. Sleep disruptions in several species of lemurs have been shown to negatively impact the performance of cognitive tasks measuring memory consolidation and foraging efficiency, as well as altering social behavior [34]. Additionally, short sleep durations and poor sleep quality are confirmed risk factors for metabolic diseases such as obesity, diabetes, and cardiovascular disease [27].

Adequate sleep is considered a fundamental factor in human wellbeing, yet is typically not taken into account in animal welfare assessments by the scientific community. Although not all focus on welfare, there are several studies on sleeping patterns in different animals, such as in laboratory rodents [35,36], a variety of farm animals [37,38], minks [39], shelter dogs [40], elephants [41], macaques [42] or lemurs [34,43]. Non-invasive studies on sleep in captive chimpanzees are scarce. To our knowledge, the first study in this field was conducted by Riss and Goodall [20], aiming to establish a connection between partner sleeping preferences and other aspects of the social relationships of a group of captive adolescent chimpanzees. However, this study was methodologically limited as it did not monitor the animals’ behavior throughout the night. During the last two decades, a few more studies have been conducted in this field. For example, Videan [15] studied the effect of several factors such as temperature, humidity, age, and sex on the quality and duration of sleep in captive chimpanzees. She found that chimpanzees do not sleep uninterruptedly, reporting frequent awakenings during the night. She also suggested that animals’ age, as well as temperature and humidity, influenced their sleep patterns. Havercamp et al. [16] focused their study on the effect of age on sleep in captive chimpanzees and found results contrary to those found by Videan [15]. To our knowledge, the only study focusing on sleep as an indicator of stress response in captive chimpanzees was conducted by Morimura et al. [44]. Here, nocturnal observations using video recordings were conducted on three chimpanzees before and after their relocation to another facility. They reported that after the relocation, chimpanzees experienced a temporary decrease in sleep duration, but eventually returned to normal sleep patterns once they adapted to their new environment. These results suggest that analysis of sleep patterns and nocturnal behavior may be a useful tool for measuring stress and wellbeing in captive chimpanzees.

Monitoring the nocturnal behavior of captive animals can be challenging due to factors such as lack of light, lack of resources, and the absence of qualified staff to personally observe the animals in indoor facilities without disturbing the animals. As a result, researchers often need to depend on electronic devices to study the animals’ nocturnal behavior. One commonly used method is to use video cameras [15,16,30,44]. However, this typically requires substantial financial investment, requires advanced technical knowledge, and can be time-consuming in the review phase [45,46]. Accelerometers have been successfully used to monitor the activity levels and body movements of non-primate animals during the day and night [38,41]. However, this method needs to be applied with care, to not negatively affect the animals themselves and is not suitable for most primate species capable of breaking or removing such devices. Although they have not been used in chimpanzee welfare research, sound recordings are commonly employed in other contexts, such as detecting the presence of species in a specific area by analyzing their vocalizations [47,48] or studying other aspects related to conservation [49]. Directly listening to the noises and vocalizations made by wild chimpanzees at night has been used in some revealing studies, describing nocturnal behavior in wild chimpanzees [10,22].

For this study, we recorded abiotic conditions (temperature, humidity) at the sleeping facilities, the occurrence and duration of nocturnal interruptions (sound events produced by chimpanzee vocalizations and behaviors), and the chimpanzee’s behavior and social proximity the following day. In regard to nocturnal interruptions, we specifically targeted sound events loud enough to effectively wake all or most of the chimpanzees residing in the same building. Analyzed sound events, which were nocturnal activities typically labeled as undesired, including noises related to displaying, drumming, fighting, and/or loud vocalizations linked to stressful, alarming, and/or aggressive events. Sleep fragmentation was represented by the frequency of such sound events, while the occurrence of high-intensity events was represented by the duration of the longest sound event each night. Both variables may serve as indicators of sleep disturbances but might produce distinct behavioral responses the following day. Nevertheless, we expect said indicators to be mutually exponentiating. Thus, we specifically focused on the combination of these indicators, including nights with few disturbances and short sound events, nights with many short disturbances, nights with few but at least one long event, and the most disturbing highly fragmented nights with long-duration events. While it may be challenging to predict differences in impact between nights in the intermediate range, it is likely that the latter will have the greatest effect on the chimpanzees’ behavior the next day. Recorded low-volume activities, such as changes in posture or location were discarded for the purpose of this study. Regarding the behavioral observations following each night, a complete ethogram has been used, yet analyses in this study focused only on behaviors potentially indicating fatigue, tension, stress, and/or social activities related to stress reduction, such as consolation or reconciliation. We established a total of three objectives, based on the data collected on two groups of captive-housed chimpanzees. Firstly, we aimed to examine if the abiotic conditions of the sleeping facilities affect the frequency and duration of night disturbances, i.e., if a warmer and more humid environment leads to more restless nights. Secondly, we expected chimpanzees’ day behavior to be affected by the frequency and intensity of sleep disruptions during the previous night. Specifically, we expected chimpanzees to be more inactive due to accumulated mental and physical fatigue; to exhibit higher levels of abnormal and self-directed behaviors (SDB), indicating stress or tension; as well as to perform more affiliative behaviors, and to increase social proximity, as a mechanism to maintain group cohesion; after more fragmented nights and/or nights containing long-lasting, intense sleep disruptions. Thirdly, we strived to demonstrate that audio recordings can be used as a cost-effective and low-tech approach, potentially providing valuable information regarding the chimpanzee night activity.

## 2. Materials and Methods

### 2.1. Study Sample

We conducted this study on two groups of former pet and entertainment chimpanzees housed at Fundació MONA, a primate rescue center located in Girona, Spain. Fundació MONA is a member of the European Alliance of Rescue Centers and Sanctuaries (EARS), which focuses on the rescue and rehabilitation of chimpanzees (*Pan troglodytes*) and barbary macaques (*Macaca sylvanus*) since 2001. The study sample consisted of 12 adult chimpanzees (5 females and 7 males), living in 2 mixed-sex groups (Appendix A). Each chimpanzee group was housed in a naturalistic outdoor enclosure (measuring 2420 m^2^ and 3220 m^2^), as well as two smaller non-naturalistic indoor areas (measuring between 25–30 m^2^ each). Outdoor enclosures consisted of a naturalistic terrain with Mediterranean vegetation (subject to seasonal changes) and were equipped with artificial climbing structures, such as towers, ropes, platforms, hammocks, ad libitum drinkers, and a variety of enrichment devices. Indoor areas consisted of cemented flooring, covered with sawdust and straw, and were equipped with platforms, climbing structures, hammocks, and ad libitum drinkers. During the colder months of the year, indoor areas were heated by an automated floor heating system. Outdoor enclosures were located beside each other, separated by a 50 m long section of steel mesh and electrified fence. All four indoor areas used by these two groups were located in the same building, sharing the same caregiver passageway. At night, the chimpanzees were confined to the indoor areas, while during the day, had either access to all areas or were restricted to the outdoor enclosures, depending on the weather conditions and maintenance/care activities. Individuals from the same group could choose the sleeping room to some extent, i.e., sleeping partners each night. Thus, each group would split its members at night time, inhabiting two indoor areas that were separated from each other (2–4 individuals per room). Chimpanzees could not see or physically interact with individuals from another room but were able to hear their vocalizations and any noisy activity, i.e., displaying and drumming. Chimpanzees were fed 4–5 times a day based on a diet consisting mostly of vegetables, fruits, pellets, and dried fruits. The first and the last meal of every day was given in the indoor areas, while the majority of their diet was scattered in the outdoor enclosures to stimulate foraging and other species typical behaviors.

### 2.2. Nocturnal Sound Recording

To record the sounds produced by the chimpanzees at night, we used a digital audio stereo recorder (TASCAM^®^ DR-07X from TEAC Corporation^©^, Tokyo, Japan) with the automatic recording function. The recording was triggered when the sound level exceeded −24 dBFS, and stopped when it remained below −24 dBFS for 5 s. We verified that this configuration effectively captured the sounds produced by chimpanzee vocalizations and activities, such as agonistic displays, agonistic conflicts, alarm calls, drumming or screaming vocalizations. The audio recorder was programmed to record 12 h per night, starting at 8:30 p.m. until at 8:30 a.m. the next day. Within this time frame, care givers would not enter these areas and artificial lighting would remain switched off. Nights with less than 12 h of recording due to animal handling or technical issues were discarded. This resulted in a total of 248 nights used in this study. In order to identify the ideal configuration of the audio recording device and to determine patterns/rules to effectively discern characteristic of sound events of interest for this study, for four months previous to the data collection, uninterrupted 12 h recordings were carefully analyzed. During this pre-analysis period, it could be observed that chimpanzees tended to produce sounds in bouts followed by periods of silence lasting several minutes (from approximately 1 to 20 min). Based on this information, we defined a sound event as a group of audio clips, with less than 20 min of silence between clips and lasting more than one minute all together. Additionally, audio clips shorter than 9 s were discarded, as they either contained irrelevant sounds without a clear chimpanzee reaction or audio artifacts produced by the device. All sound events contained at least one or several of the following items: displaying sounds, drumming, fighting, hooting, panting, pant hooting, barking, screaming, crying, whimpering, or alarm calling.

### 2.3. Measuring Abiotic Factors of Night Quarters

An indoor thermo-hygrometer was permanently installed close to the digital audio recorder, in the most central location of the chimpanzee indoor facility. The device was reset every day before closing the building and readings were taken first thing in the morning by the head caregiver. We recorded the average, minimum, and maximum values for both the temperature (°C) and the humidity levels (%) for each night.

### 2.4. Daytime Behavioral Sampling

Observations for this study were conducted from August 2021 until August 2022. Behavioral observations were conducted only while chimpanzees were located or had access to the outdoor enclosures (5640 m^2^) between 10:00 a.m. and 18:30 p.m. Observers were located on one of the two observation towers providing maximum visibility. Data were collected, using multifocal (observing all members of one group simultaneously) scan sampling, based on 2 min intervals. Each observation session lasted for 20 min, during which observers simultaneously recorded the behaviors and passive close proximity (two or more individuals remain in physical contact or up to one-armlengths reach, while not socially interacting in any way) of each chimpanzee. Data were collected using a tablet device with the behavioral monitoring app ZooMonitor [50], by a team of observers, supervised by the research coordinator of Fundación MONA. Observers were only allowed to collect data for this study if they successfully passed a three-step inter-observer reliability test. The first step included data collection for a minimum of two weeks, these data were then checked and deleted. For the second step, observers had to pass a methodology test and in the third step, they had to pass a video test, including 20 video clips, with an agreement of ≥85% to the research coordinator of the sanctuary. Scans in which the focal animal was either out of sight or its behavior was obscured were excluded. To ensure the quality of the diurnal data, we only included data from each social group on days with at least two hours of observation. This resulted in a total of 610 h of observations distributed over 133 days (279 ± 104 h per individual). The ethogram used is included in the accompanying Appendix A.

### 2.5. Statistical Analysis

#### 2.5.1. Correlation between Abiotic Factors and Nocturnal Events

To explore potential influences of abiotic factors on the chimpanzee’s nocturnal activity, i.e., sleep interruptions, Kendall’s rank correlation tests (adjusted *p* values with Holm–Bonferroni method) were conducted, after confirming the non-normal distribution of the variables. Variables regarding the abiotic factors were the average, maximum, and minimum relative humidity and temperature values recorded at the night quarters. Variables representing the nocturnal activity were the number of sound events during the night (frequency) and the duration of the longest event of each night (intensity). Consult Appendix A for more detailed information regarding those variables.

#### 2.5.2. Linear Mixed Models

To examine the potential impact of nocturnal disruptions on the daytime behavior of the following day, we ran three linear mixed models (LMMs) with “Inactivity”, “Self-directed & Abnormal behaviors” and “Social Proximity & Affiliative behaviors” as respective dependent variables. Fixed factors were the same in all LMMs, only the dependent variable differed for each model. We included a total of four fixed factors in our full models. The fixed factors representing the sleep disruptions were established based on the number of validated sound events (N° events: one vs. two vs. three or more) and the duration of the longest validated sound event (max duration: short vs. medium vs. long) of each night. More detailed information regarding the ranges of each category can be found in Appendix A. Furthermore, we added the chimpanzee’s sex (male vs. female) and age category (adult vs. senior 30+) as fixed factors in all models; and used the chimpanzee’s ID and group as random factors (nested). We visually checked QQ plots for a normal distribution of the residuals (Appendix A). We tested for multicollinearity between all fixed factors by calculating the variance inflation factor (VIF) using the “car” package in R [51]. All VIFs (variance inflation factor), calculated for our three fixed factors were below 1.03, indicating that our fixed factors were not correlated. First, we tested whether full models (containing all four fixed factors) were significant improvements over the null models (without fixed factors). In case a full model differed significantly from the corresponding null model, we applied the ANOVA function (Type III Analysis of Variance with Satterthwaite’s method) and a post hoc test based on the *p*-value obtained with the “glht” function (multiple comparison of means with Tukey contrast, *p*-values adjusted by the Holm–Bonferroni method). Two of the three models were conducted several times, after including the interactions between previously found significant fixed factors, and conducted post hoc tests based on the *p*-value obtained with the “emmeans” function (adjusted *p* values with Bonferroni method). All of the models were run using the “lme4” package [52] in R version 4.2.0 (R Foundation for Statistical Computing, Vienna, Austria) [53].

## 3. Results

### 3.1. Correlation between Abiotic Factors and Nocturnal Events

A total of twelve correlations between the six variables regarding the abiotic factors (T^a^, T_min_, T_max_, H^a^, H_min_, H_max_) and the two variables regarding the nocturnal event indicators (N° events, max duration) were conducted (Appendix A). Out of these twelve tests, seven produced significant results. We found that N° events were positively correlated with all three temperature-related variables (T^a^: Z = 5.6, *p* < 0.001, tau = 0.27; T_min_: Z = 6.1, *p* < 0.001, tau = 0.3; T_max_: Z = 4.9, *p* < 0.001, tau = 0.24), meaning that significantly more events occurred at nights marked by higher temperatures. The N° events were negatively correlated with the maximum relative humidity in the indoor enclosures (Z = −2.8, *p* = 0.005, tau = −0.14), meaning that fewer events occurred during nights marked with high H_max_ values. Lastly, max duration was positively correlated with all three temperature-related variables (Z = 2.7, *p* = 0.007, tau = 0.12; T_min_: Z = 2.8, *p* = 0.005, tau = 0.12; T_max_: Z = 2.6, *p* = 0.009, tau = 0.11), meaning that the duration of the longest event was longer at nights marked by higher temperatures.

### 3.2. Linear Mixed Models

We calculated three different behavioral variables, based on data collected during daytime observations following each monitored night, i.e., time spent on “Inactivity”, “Self-directed & Abnormal behaviors” and “Social Proximity & Affiliative behaviors” and ran LMMs to investigate the effects of nocturnal disruptions (N° events, max duration) and the chimpanzee’s biographic information (sex, age category). We found all three full models to show significant improvements compared to their respective null models. Sex was the only fixed factor to never produce any significant results in any of the full models. All outcomes of the three LMMs and the respective post hoc analysis are presented in Appendix A.

#### 3.2.1. Impact on the Occurrence of Inactivity

The full model with inactivity behavior as the dependent variable revealed a significant effect on the number of events (F = 26.3382, *p* < 0.001), the duration of the longest event of the night (F = 19.6885, *p* < 0.001), and the age category (F = 6.1186, *p* = 0.02924). In relation to the N° events, we found that the higher the number of events, the more inactivity was recorded the next day. Significant differences were observed between one and two events (one < two; Z = 4.720, *p* < 0.001), one and more than three (one < three or more; Z = 7.141, *p* < 0.001), and between two and more than three (two < three or more; Z = 2.476, *p* = 0.0133). Regarding max duration, we found that when nighttime events lasted longer, inactivity levels also generally increased. Significant differences were observed between short and medium (short > medium; Z = −2.144, *p* = 0.032), between short and long (short < long; Z = 6.189, *p* < 0.001), and medium and long sound events (medium < long; Z = 4.079, *p* < 0.001). With respect to the age category, we found adult chimpanzees to perform fewer inactivity behaviors than seniors (adult < senior; Z = 2.474, *p* = 0.0134). Statistical results of the “inactivity” model and the respective post hoc analysis are presented in Appendix A; confidence interval plots regarding the significant fixed factors are shown in Figure 1.

Considering that both N° events and max duration significantly affected inactivity levels the following day, we performed a separate model based on the “inactivity” full model, now including the interaction of these two variables. We found said interaction to have a significant impact on inactivity (F = 5.4806, *p* < 0.05). Post hoc test results are presented in Appendix A; a graphical representation of said interaction between significant fixed factors is shown in Figure 2. While considering the number of events, nights marked with a long-lasting disruption were always followed by higher levels of inactivity compared to nights marked with short or medium disruptions as well as positively correlated with rising N° of events. Nights marked by a short night disruption were also positively correlated with rising N° of night events, but generally scored lower levels of inactivity compared to nights marked with a long-lasting disruption. However, the mid-range nights marked as medium-lasting night disruption did not follow the expected course, as the increase between “one” and “two” was less steep than seen in “short” (max duration) and no significant change between “two” and “three or more” (N° events) occurred.

We conducted separate models including the interaction of N° events and max duration with age category, respectively, in order to investigate inactivity changes within and between each Age Category depending on the night disturbances. We found the interaction of max duration and age category to not have any significant impact on inactivity, while the interaction between the N° of events and the age category did have a significant impact on inactivity (F = 3.1158, *p* < 0.05). The graphical representation (Figure 3) indicates that seniors always scored higher levels of inactivity even without considering N° events. Both age categories increased levels of inactivity with increasing N° of events. Furthermore, the trendline of seniors is showing a slightly steeper increase in inactivity with an increasing number of night events, when compared to adults. However, post hoc tests based on the comparison of means only produces significant results between N° of events categories within the same age category, but not between age categories (Appendix A).

#### 3.2.2. Impact on the Occurrence of Self-Directed and Abnormal Behaviors

The full model with self-directed and abnormal behaviors as the dependent variable revealed a significant effect of max duration (F = 15.7027, *p* < 0.001). We found that when nighttime sound events were labeled as “long”, self-directed and abnormal behaviors were significantly higher the next day. No significant differences are observed between short and medium (Z = 0.209, *p* = 0.834), but significant differences were found between short and long (short < long; Z = 4.978, *p* < 0.001) and medium and long (medium < long; Z = 4.772, *p* < 0.001). Statistical results of the “Self-directed & Abnormal behavior” model and the respective post hoc analysis are presented in Appendix A; confidence interval plot regarding the significant fixed factor is shown in Figure 4.

#### 3.2.3. Impact on the Occurrence of Social Proximity and Affiliative Behaviors

The full model with social proximity and affiliative behaviors as the dependent variable revealed a significant effect on the number of events (F = 15.6418, *p* < 0.001) and the duration of the longest event of the night (F = 18.3566, *p* < 0.001). In relation to N° events, we found that the more events at night, the more social proximity and affiliative behaviors were recorded the next day. Significant differences were observed between 1 and 2 events (one < two; Z = 4.210, *p* < 0.001) and 1 and more than 3 (one < three or more; Z = 5.310, *p* < 0.001). In regard to max duration, we found that when nights were labeled as “long”, social proximity and affiliative behaviors occurred significantly more often. Significant differences were found between short and long (short < long; Z = 5.581, *p* < 0.001) and medium and long (medium < long; Z = 4.866, *p* < 0.001). Statistical results of the “Self-directed & Abnormal behavior” model and the respective post hoc analysis are presented in Appendix A; confidence interval plots regarding the significant fixed factors are shown in Figure 5.

Considering that both N° events and max duration significantly affected “Social proximity & Affiliative behavior” occurrences the following day, we performed a separate model based on the “Social proximity & Affiliative behavior” full model, now including the interaction of these two variables. We found said interaction to have a significant impact on the time spent in “Social proximity & Affiliative behavior” (F = 2.4149, *p* = 0.05). Post hoc test results are presented in Appendix A; a graphical representation of said interaction is shown in Figure 6. The plot shows as already indicated in the original “Social proximity & Affiliative behavior” model, that (a) nights with more sound events are followed by days with higher levels of social proximity and affiliative behavior and (b) nights containing max duration events labeled as “long” are also followed by days with higher levels of social proximity and affiliative behaviors. The interaction plot, however, also shows that nights containing max duration events labeled as “long” are generally always followed by higher levels of “Social proximity & Affiliative behavior”, in comparison to nights labeled as “short” and “medium”, regardless the level of N° events. Furthermore, nights labeled as “long” present a clear positive correlated tendency towards higher occurrences of social proximity and affiliative behavior, while increasing N° event levels. However, max duration events labeled as “short” and “medium” only show this tendency between the N° events labeled as “one” and “two”, but no significant differences were found between “two” and “three or more”.

## 4. Discussion

In this study, we examined the potential effect of indoor abiotic conditions (temperature and humidity) on the sleep disruption of chimpanzees by analyzing the frequency and longest duration of loud nocturnal sound events, recorded with a digital stereo audio recorder. Specifically, we recorded and analyzed sound events, loud enough to ensure all chimpanzees, housed in the same building would wake up. Said sound events were nocturnal activities, typically labeled as undesired, if occurring frequently, such as displaying, drumming, fighting, and/or loud vocalizations typically indicating stress, alarm, and/or aggressions. Our results demonstrated a positive correlation between the number and maximum duration of sound events and the average nighttime temperature in indoor facilities. These findings are consistent with those of Videan [15], who observed that higher temperatures decreased the sleep quality of the chimpanzees in their study. Surprisingly, that maximum humidity correlated negatively with the number of sound events, contradicting the results of Videan [15], where humidity correlated negatively with sleep quality. These discrepancies may be due to differences in the methods and definitions of indicators used to measure sleep quality. Furthermore, we are aware that, while all variables related to temperature (T^a^, T_min_, T_max_) correlated positively with N° events and max duration; H_max_ was the only variable regarding humidity to produce a significant result. Videan [15] used video recordings to collect data and analyzed chimpanzee sleep patterns by observing body movements and postures. In our study we used audio recordings; focusing on the occurrence of loud noises produced by the chimpanzees themselves. While the presence of loud sounds, produced by chimpanzees themselves, may serve as an indication of unsolicited sleep disruption, the absence of such sounds does not necessarily mean that the chimpanzees are experiencing undisturbed sleep. Night activity itself should not be seen as something negative or undesired in general, as chimpanzees in the wild were also observed to be active to some degree at night [18,19,20,21] and captive chimpanzees tend to be restricted for relatively long hours in night areas. Rather it seems important to differentiate between quiet, neutral, or even amicable night activities (unlikely to disturb or wake other members of the population) and noisy or even hostile activities which force all or most chimpanzees housed in the same facility (but not necessarily same quarter) to wake and to be placed in alert and/or a social conflict. That being said, our results suggest that certain abiotic conditions, such as higher average indoor temperatures and lower humidity levels, seem to encourage chimpanzees to exhibit noisy nocturnal behaviors, which may in turn negatively impact the sleep quality of group members and others close by populations. This interpretation is supported by the findings of Piel [22], who observed more frequent vocalizations at night in a wild chimpanzee population in Tanzania when the temperature was higher and relative humidity was lower. However, by no means do we intend to suggest temperature and humidity be the only factors impacting night activity. These variables were chosen because they are potentially easily controlled and monitored by care staff and are likely to fluctuate along the year, thus were likely to offer a range that could be analyzed in order to draw conclusions within the one-year data collection time frame of this study.

Living conditions in indoor areas as well as the impact of noisy nocturnal activity on the next day’s behavior of diurnal animals is an understudied topic, typically not taken into account when assessing animal welfare in captivity. A recent study by Schork et al. [30] investigated the effect of daytime activity on subsequent nocturnal activity and the effect of nocturnal activity on the next day’s diurnal activity in laboratory-housed dogs. The findings in this study are consistent with those of our study on sanctuary chimpanzees. In both studies, an increase in nocturnal sleep fragmentation was associated with an increase in inactivity behaviors the next day. In this study, other than sleep fragmentation we found the intensity of the longest sound event to also impact the exhibition of inactivity. This was expected as not getting enough quality sleep tends to produce an accumulation of mental and physical fatigue. On the contrary, a certain level of sleep quality is essential to keep energy levels up, as well as increase attention and concentration capacities [54,55].

However, it is important to point out, that both variables representing night disruptions were analyzed by pooling the frequency of events and the duration of the longest events into categories. The conversion of continuous variables into categorical variables always comes with the risk of losing details to some degree. Nevertheless, it was necessary to implement categories, as this allowed us to guarantee a minimum amount of behavioral data as well as even distribution of observations throughout the hours of the day, to objectively reflect the behaviors of the days following certain night conditions. Establishing the ideal range for such categories ideally requires a large amount of data and needs to be completed carefully. Regarding the categories established for this study, we found in both variables that extreme values, i.e., “short” and “long” and “one” and “three or more” followed the expected tendencies, providing clear significant results, yet middle range values, i.e., “medium” and “two” were less consistent. The most likely explanation is that middle-range categories need to be redefined, by either rearranging all categories or reducing these variables to two-level categorical variables. However, we believe that future analysis based on continuous variables will enable us to establish more objective and representative categories, once the database could be sufficiently expanded.

That being said, our results suggest that nights with more frequent and/or long-lasting sound events lead to higher levels of lingering social stress among individuals in the group. Considering that the recorded sounds in this study typically contained noises produced by behaviors and vocalizations indicating stressful, alarming, or aggressive events, the observed increase in affiliative behaviors and social proximity might be motivated by attempts to console or reconcile with group members the following day. All the longest sound events used to extract information for the max duration variable that were labeled as “long” included clear indications of intense social conflicts. Chimpanzees are known for their capacity to navigate complex social situations and to use peaceful post-conflict interactions [56] to reduce the costs of aggressive escalations [57]. Although grooming is likely to be the most used social behavior to console and/or reconcile, Webb et al. [58] demonstrated recently that chimpanzee reconciliation patterns and preferences are highly individualized and besides grooming also include behaviors such as kissing, embrace, touch, finger/hand in mouth, play and even passive close proximity or sitting in contact. Another explanation might be that intense night disruptions might lead to an increase in agonistic events the following day which in continuation might have influenced the observed increase in abnormal and self-directed behaviors, affiliative behaviors, and social proximity. However, agonistic events during the day were rarely recorded due to their seldom occurrence and often lacked the necessary duration to be efficiently captured by the 2 min interval data collection method. Thus, we were not able to confirm this possibility based on our data set. Self-directed behaviors in primates and other mammals are often used as indicators of tension and negative emotions [59]. SDBs have been associated with stress and frustration in non-social contexts [60] as well as in social contexts, such as post-conflict situations [61]. Several studies in humans also demonstrated that poorer sleep quality is associated with lower emotion-regulation ability [62,63], increases stress levels and may even lead to depression [64]. The occurrence of abnormal behaviors is a more controversial indicator of stress and welfare. Although traditionally considered reliable indicators of psychological distress and poor welfare [65], more recent research suggests that not all behaviors considered abnormal have the same etiology and should not be analyzed together [66]. Furthermore, the expression of abnormal behaviors in captive primates shows a large variation among individuals, both in frequency and duration [67]. Recently, a study by Goldborough et al. [68] investigating qualitative individual variations in abnormal behaviors of chimpanzees, found considerable variations in terms of diversity and overall composition. Furthermore, their results indicate that these variations were not influenced by biographic factors such as age, sex, or hierarchy position within the group. Our results support these findings as neither sex nor age influenced the exhibition of abnormal behaviors. Furthermore, although we did not differentiate between different abnormal behaviors, nor conducted detailed individual analysis, differences in abnormal diversity could be clearly seen as well as standard deviations of observed abnormal and self-directed behaviors also suggest major variations between individuals in our study population. Nevertheless, night disturbances labeled as particularly long seemed to affect all chimpanzees in a similar way, i.e., produced an increase in said behavior the next day. Although we agree with those recent viewpoints regarding abnormal behaviors, within the framework of this study we pooled the occurrences of abnormal and self-directed behaviors into a single variable, arguing that the fluctuation and temporary increase in both behaviors are still likely to function to some degree as an indicator of accumulated stress and tension. Moreover, the majority of the abnormal behaviors observed in this study were behaviors generally labeled as “self-directed”, but exhibited in excessive or unfunctional manner, such as over-grooming. As previously mentioned, it would be highly valuable for future studies with larger datasets to differentiate between specific abnormal behaviors and individual variations, in order to better understand the different coping mechanisms that chimpanzees employ in response to social stress caused by conflicts, whether they occur during the night or day.

Regarding the biographic factors, the chimpanzee’s sex did not influence any of the behavioral variables, while the age category only affected the exhibition of inactivity. As expected, seniors were generally observed to spend more time in inactivity when compared to younger adult chimpanzees, supporting the results of other studies on this subject [69]. Moreover, our data indicate that senior chimpanzees might be slightly more strongly impacted by more fragmented nights in comparison to their younger counterparts. However, the latter is only based on an observed tendency, which could not be confirmed by post hoc test results, as well as would contrast results from previous studies on sleep quality in aged chimpanzees [15,16]. Hence, a bigger sample population is required to provide clearer results.

As mentioned before, no infants or juveniles were present in our study population, which limits or capacity to discuss the impact of night disturbances in regard to the chimpanzees age. Hence, we could not provide any information on how night disturbances may impact younger chimpanzees in comparison to adults and seniors. Furthermore, the presence of restless infants and energetic juveniles might produce additional night disturbances that could potentially affect the chimpanzee’s behavior the next day in similar ways as discussed in our results. However, in order to establish such a conclusion, we need to expand our data collection to include more populations with varying age ratios.

## 5. Conclusions

According to the results obtained in this study, the temperature and humidity levels at the chimpanzee’s night quarters do influence their nocturnal activity, i.e., frequency and intensity of sound events. Furthermore, said sleep disruptions have a significant impact on chimpanzee behavior the following day. Specifically, we found that after nights with frequent and/or intense interruptions, chimpanzees were observed the following day to increase time spent on inactivity, SBD, and abnormal behaviors, as well as spend more time in social affiliative interactions and reside in close proximity to others. These results highlight the importance of providing adequate night quarters while controlling and monitoring abiotic factors that could negatively affect the animal’s sleep quality and/or produce undesired nocturnal activity in captive-housed chimpanzees. It is important to state here, that sound events recorded and analyzed in this study, consisted of noisy and typically undesired behaviors such as displaying, drumming, and fighting and/or loud vocalizations typically indicating stress, alarm, and/or aggressions (although no serious night wounding, requiring veterinary attention has been reported throughout the study). Thus, calm and inoffensive nocturnal activity, such as changes in posture, behaviors related to alimentation or affiliative interactions would not produce the necessary volume to wake all chimpanzees in the building, hence has not been recorded due to the configuration of the recording device.

Although much more information and research are still required, we wish to highlight the importance to dedicate resources and increase our knowledge regarding best practice indoor (night quarter) creation. As these facility sections tend to be less monitored by care staff and often remain out of sight of visitors and observers, they tend to be neglected in comparison to typically larger, better equipped, and enriched outdoor enclosures. Yet, it is important to remember, that in captivity, animals might spend the same or even more hours of the day in these indoor areas (depending on the organization and climatic conditions). Hence, facilitating nighttime enclosures promoting individual wellbeing is of utmost importance and may require some type of long-term monitoring mechanism. Furthermore, we believe that information regarding the extent of undesired sleep interruptions during the nights can function as a useful tool for care-management decisions as it potentially allows us to predict behavioral tendencies of the next day.

In order to strengthen the findings discussed in this study, more data needs to be collected and the study sample needs to be increased, ideally including populations housed in other organizations, using different types of night quarters which would allow increasing the list of potential factors influencing the chimpanzees sleep quality (for example habitat size, square meters per chimp, nesting material, the total amount of conspecifics in the audible distance, etc.). Nevertheless, our data already indicates that the abiotic factors of the night quarters affect the chimpanzee’s restlessness as well as demonstrates that frequently interrupted nights and/or nights containing long and intense sound events do affect the chimpanzee’s activity budgets the next day. We are hesitant in regard to suggesting definite values of “ideal” temperature and humidity, as our results are based solely on one study population and best practice manuals established by organizations such as AZA, EAZA, PASA, or NAPSA already provide such values. However, based on our findings, we suggest that one way to reduce the high nocturnal activity of captive chimpanzees might be achieved by decreasing the temperature within the range advised by these best practice guidelines.

Furthermore, we managed to demonstrate that economically affordable equipment, such as the here applied audio recordings, has the potential to provide valuable information on chimpanzee nocturnal behavior and wellbeing and can be used to efficiently monitor nocturnal activity. Although video recordings are likely to provide more detailed insights, the necessary technical equipment tends to be much more expensive and technical skills might need to be more advanced. Thus, the efficiency and balance between practicality and resource requirement are likely to be an issue for many primate housing organizations.

## Figures and Tables

**Figure 1 animals-13-00696-f001:**
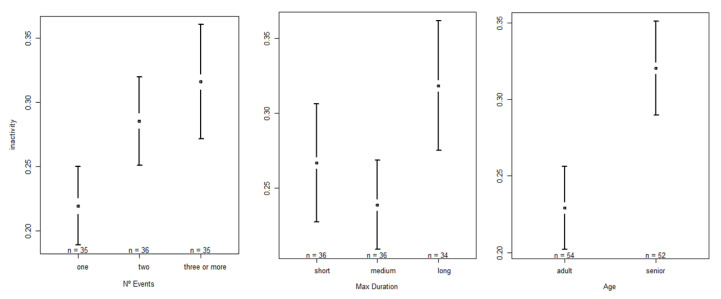
Confidence interval plots of inactivity behavior and the three significant fixed effects: number of sound events at night (N° events), duration of the longest event (max duration), and age category.

**Figure 2 animals-13-00696-f002:**
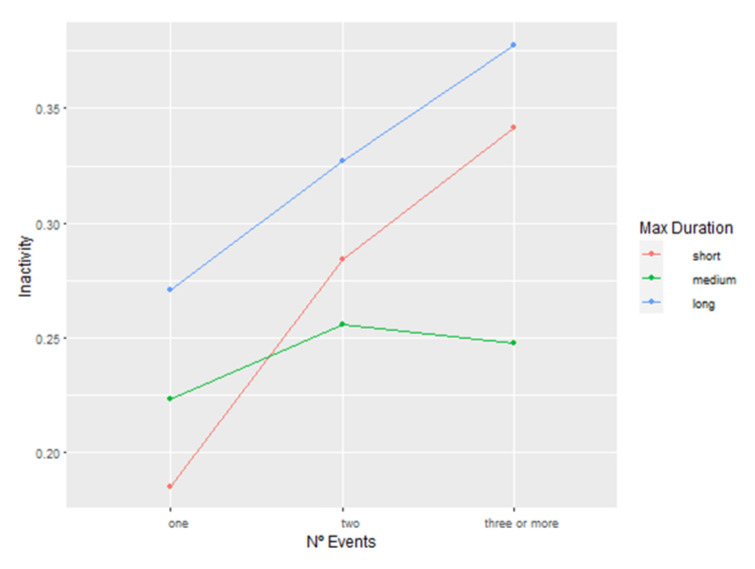
Plot representing the effect of the interaction between N° events and max duration on daytime inactivity of the chimpanzees. Each point on the plot is a predicted mean inactivity value and each connection between two points describes the effect, based on the data of the inactivity LMM model with N° events, max duration, sex, age category, and the interaction of N° events, max duration as fixed factors.

**Figure 3 animals-13-00696-f003:**
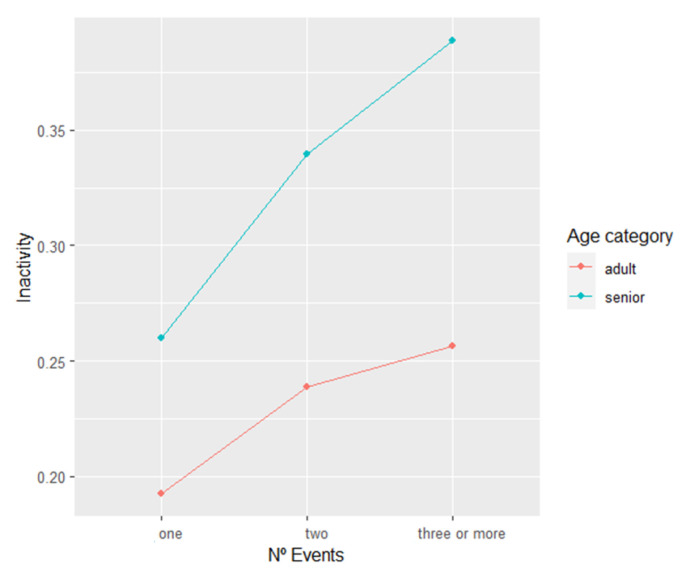
Plots representing the effect of the interaction between age category and N° events on daytime inactivity of the chimpanzees. Each point on the plot is a predicted mean inactivity value and each connection between two points describes the effect, based on the data of the Inactivity LMM model with N° events, max duration, sex, age category, and the interaction of age category and N° events as fixed factors.

**Figure 4 animals-13-00696-f004:**
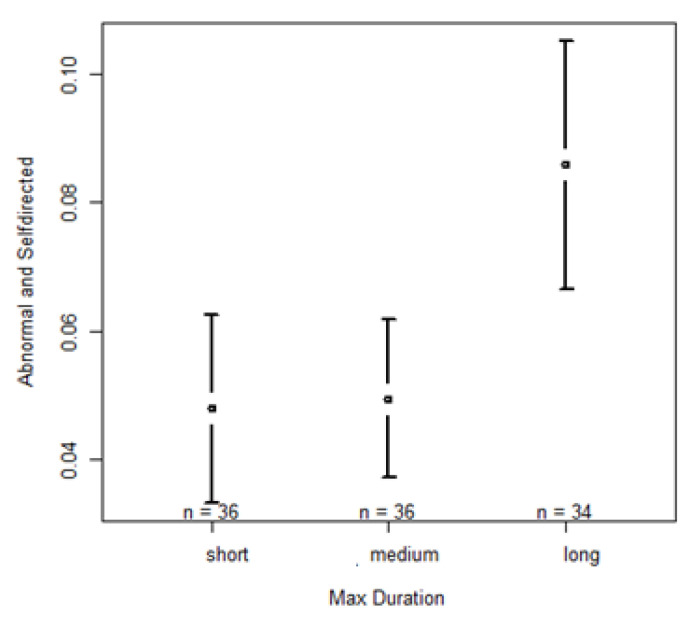
Confidence interval plots of abnormal and self-directed behaviors and the fixed effect duration of the longest event (max duration).

**Figure 5 animals-13-00696-f005:**
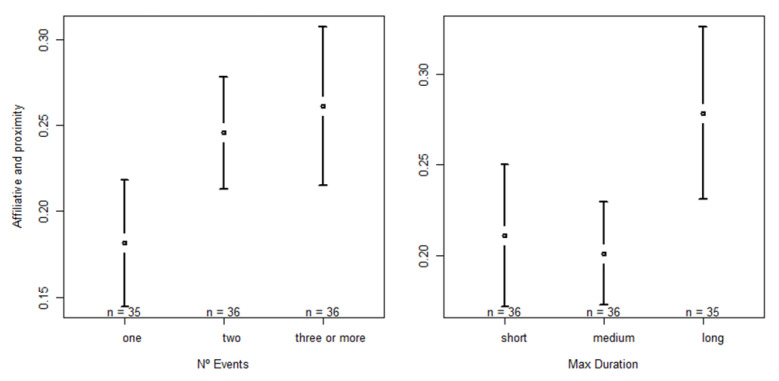
Confidence interval plots of social proximity and affiliative behaviors and the fixed effects number of sound events at night (N° events) and duration of the longest event (max duration).

**Figure 6 animals-13-00696-f006:**
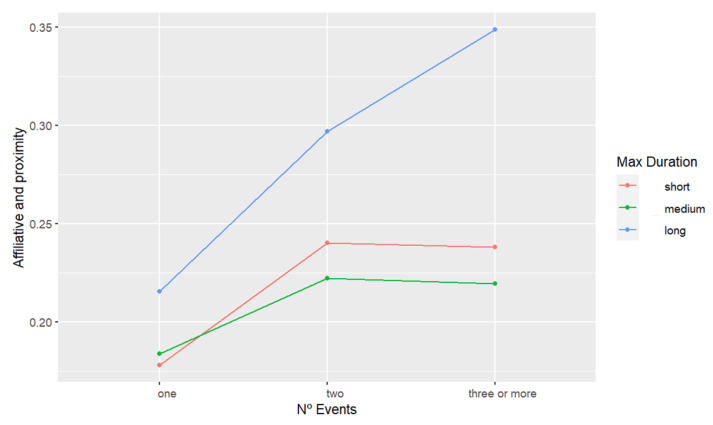
Plot representing the effect of the interaction between N° events and max duration on daytime affiliative and proximity behaviors of the chimpanzees. Each point on the plot is a predicted mean “Social proximity & Affiliative behavior” value and each connection between two points describes the effect, based on the data of the “Social proximity & Affiliative behavior” LMM model with N° events, max duration, sex, age category, and the interaction of N° events, max duration as fixed factors.

## Data Availability

Data are contained within the Appendix A.

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
