# Peer review of "Listening to Their Nights: Sleep Disruptions in Captive Housed Chimpanzees Affect Their Daytime Behavior"

_animals, 2023, doi:10.3390/ani13040696_

Round 1
Reviewer 1 Report
Please see the attached word file.

Author Response
We would like to express our sincere gratitude for all the corrections and suggestions you have provided. We are especially grateful for the promptness of your response and the dedication you have shown to improve our manuscript. We attach the responded version of your review in pdf format. Thank you very much.

Reviewer 2 Report
Basic Reporting
This study investigated how sensitive chimpanzees are towards sleep quality. The authors examined how environmental factors, like temperature and humidity and disturbance affects nighttime sleep, which in turn has consequences for chimpanzee behaviors the next day.
This is an interesting and timely topic. The existing knowledge on the effects of sleep and sleep deprivation in non-human animals is very limited. I also particularly endorse the authors attempt to investigate the effect of sleep interruption on chimpanzee welfare. Such knowledge can deliver valuable insights for captive management policies.
Detailed comments:
I just wondered why not the real duration was used in the models as measurement of intensity. This would make for more fine scaled data compared to the categories: short, medium and long. Also, it would be interesting if the authors could let us know whether they tested for any potential interaction effect between sound intensity and sound duration during nighttime interruptions?
I am aware that this may not be possible without extensive sound analysis, yet it would be interesting as a follow up to explore whether the social behaviours the next day somehow relates to who was giving agonistic vocal signals in the night. Of course, this would require identity assessment possibilities from each call from the nighttime sound recordings.
I find it quite interesting that social affiliative behaviours increased after noisy nights, despite that this was when temperatures were higher, opposite to the alternative explanations that close social proximity is beneficial when it is colder. As such, these results really suggests that they spend time to maintain their relationships after disturbance or occasions of arousal. Thus, it would be worth looking into the details on who is pending time with whom the day after especially “high arousal” nights.
I am very much impressed by this very nicely planned and conducted study and do not have much to add.
Author Response
Basic Reporting
This study investigated how sensitive chimpanzees are towards sleep quality. The authors examined how environmental factors, like temperature and humidity and disturbance affects nighttime sleep, which in turn has consequences for chimpanzee behaviors the next day.
This is an interesting and timely topic. The existing knowledge on the effects of sleep and sleep deprivation in non-human animals is very limited. I also particularly endorse the authors attempt to investigate the effect of sleep interruption on chimpanzee welfare. Such knowledge can deliver valuable insights for captive management policies.
Detailed comments:
I just wondered why not the real duration was used in the models as measurement of intensity. This would make for more fine scaled data compared to the categories: short, medium and long.
Authors' reply: "Thank you for your feedback. The decision to use the longest event duration as a measure of sleep disruption intensity at night was made to maintain clarity in detecting high intensity agonistic or alarm episodes associated with long sound events (over 19 minutes in the current study). Using other variables, such as total duration, could result in nights with numerous short sound events being perceived as having the same intensity as nights with a single long event. Since events of different durations seem to be of different nature, we chose to use the duration of the longest event to differentiate between more or less intense events. This approach allowed us to detect high intensity contexts without the potential uncertainty introduced by alternative variables. The main objective of this project is to develop a nocturnal monitoring protocol that can reliably detect high stress situations in animals, whether social or not, without the need for constant manual evaluation of data nor the need to use complicated and/or costly audio analysis software. This protocol is supposed to serve as an early warning system, enabling the animal care team to review relevant audio or video recordings for informed decision making in the management of chimpanzees."
Also, it would be interesting if the authors could let us know whether they tested for any potential interaction effect between sound intensity and sound duration during nighttime interruptions?
Authors' reply: "Thank you for bringing this highly relevant issue to our attention. During the pre-analysis phase, it was observed that nights with long audio clips were associated with more intense social conflicts than nights with shorter audio clips. In future developments of this project, we aim to more accurately classify the relationship between intense agonistic contexts and event duration by incorporating a video observation phase, which will enable us to further refine this relationship."
I am aware that this may not be possible without extensive sound analysis, yet it would be interesting as a follow up to explore whether the social behaviours the next day somehow relates to who was giving agonistic vocal signals in the night. Of course, this would require identity assessment possibilities from each call from the nighttime sound recordings.
Authors' reply: "This is a highly intriguing aspect that could provide us with valuable information for informing animal care decisions. With the current methodologies used in our work, this type of analysis you suggest is not feasible. Although care givers were consulted to identify the chimpanzees emitting the noises and vocalizations, by only using audio recordings we had no means to objectively verify this information. However, we are currently exploring the possibility of using more advanced technologies, such as artificial intelligence, to facilitate the detection of additional details, such as the exact location within the facility and the emitter of the vocalizations, in a time-efficient manner. We are still in the early stages of the chimpanzee nocturnal activity monitoring project, and we plan to continue developing this project and to show our progress in future publications.”
I find it quite interesting that social affiliative behaviours increased after noisy nights, despite that this was when temperatures were higher, opposite to the alternative explanations that close social proximity is beneficial when it is colder. As such, these results really suggests that they spend time to maintain their relationships after disturbance or occasions of arousal. Thus, it would be worth looking into the details on who is pending time with whom the day after especially “high arousal” nights.
Authors' reply: "Thank you for this proposal. We definitely agree that this would be very useful information and would provide insights helping us understand how different social strategies emerge in individual animals after high-intensity nocturnal events and how these events affect inter-individual relationships. At our facilities, we frequently conduct projects using social network analysis and we hope at some point to integrate night time activity with following day time activity in our social networks. We are very grateful for your suggestion and will keep working on this topic."
I am very much impressed by this very nicely planned and conducted study and do not have much to add.
Authors' reply: "We are extremely grateful for the positive opinion regarding the usefulness and importance of the study topic presented. Thank you very much for providing us with suggestions to further improve this and the following projects regarding the chimpanzees night life."
Round 2
Reviewer 1 Report
I want to express my admiration and gratitude for the clear and complete manner in which the authors responded to all comments and suggestions. The improved manuscript maintains all the good qualities from the original and further elaborates where matters were previously unclear. Especially the results section is much clearer now, and I am happy to see that the additional analysis on the interaction of age and disturbance was implemented and found interesting results that fit well with the rest of the study.
I see no further issues or suggestions. It is evident that the authors have been very thorough in their conception and implementation of this study, and in the new manuscript this should also be clear to the readers. I am glad that my review was of assistance in the discussion of this interesting topic.